# Optimization of Printed Polyaniline Composites for Gas Sensing Applications

**DOI:** 10.3390/s22145379

**Published:** 2022-07-19

**Authors:** Ciril Reiner-Rozman, Bernhard Pichler, Vivien Madi, Petra Weißenböck, Thomas Hegedüs, Patrik Aspermair, Johannes Bintinger

**Affiliations:** 1Department of Physics and Chemistry of Materials, Danube Private University, 3500 Krems, Austria; 2Biosensor Technologies, Austrian Institute of Technology, 3430 Tulln, Austria; bernhard.pichler@ait.ac.at (B.P.); vivien.madi@ait.ac.at (V.M.); thomas.hegedues@ait.ac.at (T.H.); patrik.aspermair@ait.ac.at (P.A.); 3Department of Chemistry, University of Vienna, 1090 Vienna, Austria; a01505218@unet.univie.ac.at

**Keywords:** gas sensing, polymer doping, electrical smell sensing, biomarker detection, electronic nose, breath cancer diagnosis, principal component analysis for gas discrimination

## Abstract

Polyaniline (PANI) films are promising candidates for electronic nose-based IoT applications, but device performances are influenced by fabrication parameters and ambient conditions. Affinities of different PANI composites to analytes for gas sensing applications remain elusive. In this study, we investigate the material properties in detail for two different dopant systems: F4TCNQ and carbon black. Using a reproducibility-driven approach, we investigate different dopant concentrations in regard to their sensitivity and specificity towards five relevant markers for breath cancer diagnosis. We benchmark the system using ammonia measurements and evaluate limits of detection. Furthermore, we provide statistical analysis on reproducibility and pave the way towards machine learning discrimination via principal component analysis. The influence of relative humidity on sensor hysteresis is also investigated. We find that F4TCNQ-doped PANI films show improved reproducibility compared to carbon black-doped films. We establish and quantify a tradeoff between sensitivity, reproducibility, and environmental stability by the choice of dopant and concentrations ratios.

## 1. Introduction

Investigations of electronic nose sensor arrays began in 1960 [1]. Most commonly, metal oxide (MOx) sensor systems are used for environmental gas sensor monitoring or the measurement of total volatile organic compounds (VOC). Although these sensor systems are ideal candidates for certain applications, conductive polymer (CP) sensor systems show very limited sensitivity towards small gas molecules (e.g., CO, H, and COOH) [2], which can be an advantage when specifically aiming for the detection of certain VOCs and/or their combinations. Furthermore, the key technological advantages of using polymers instead of metal oxides are room temperature operation, low production costs, and a wide range of available materials with different specificities for target analytes [3]. Conductive polymers are widely used in organic electronics applications, such as displays, capacitors, photovoltaics, and bio-sensors. Improving their electrical performance and properties through doping strategies brought them much attention over recent years and can broaden the use of suitable materials for electronic nose (eNose) arrays.

Chen et al., in 2002, demonstrated sensitivity improvements of polyaniline (PANI) films doped with carbon black filler materials towards BTX components, ammonia, and more [4]. PANI and the discovery of the conductivity of the emeraldine salt form led to great interest in this polymer. It occurs in three oxidation states, namely as fully reduced leucoemeraldine, semi-oxidized emeraldine, and fully oxidized pernigraniline base [5]. PANI composites for gas sensing were proposed in a variety of studies [6,7,8], including nanoribbon [9] and other structures utilizing sophisticated equipment. Other approaches included PANI films in sensor arrays of a variety of polymeric materials forming e-Nose sensing systems for the discrimination of e.g., meat freshness [10], kidney disease [11], or candies [12]. Technological advances from the last centuries significantly impacted the facilitation of a polymer-based sensor system. Shirakawa’s Nobel prize work in 2000 [13] (enabling chemical modifications for polymer conductivity, which allows for pi-conjugated materials using different functional groups within the pi-system but also their corresponding side-chain modifications) opened a plethora of new organic electronic materials to be used and optimized for tailormade applications such as eNoses (e.g., increase in specificity). Recent progress in the microfabrication of electronic circuit boards [14,15] provides smaller electrode distance patterns, and hence increases the reproducibility and scalability of chemiresistor sensors. Advances in machine learning and artificial intelligence [16,17] are providing a perfectly suitable route for data evaluation of eNose response data, hence increasing the applicability of chemiresistor sensor approaches by optimized output handling. Advances in printing technology allow for high precision printing of micrometer scale polymer dots and the use of polymer solutions with low limitation to solution properties [3]. Recent work shows the state-of-the-art progress in this direction [18]. Still, commonly used polymer materials exhibit a broad specificity range towards several analytes rather than just one. Hence, several different conductive polymers with different but overlapping specificities need to be combined in order to achieve more powerful eNose systems. PANI composites show sensitivity towards a variety of compounds e.g., hydrogen sulfide [19], xylenes [20], and NO_2_ [21]. The aim of our study is the realization of cost-efficient sensor units, suitable for IoT applications, with low energy consumption, scalable fabrication methods, and efficient preparation of the functional materials needed. In previous studies, our team showed the possible routine for fabrication of sensitive but inexpensive sensor units [22,23,24]. Due to the compatibility of the chemiresistive technology to Internet of Things applications as a result of their room temperature operation, real-time read-out and energy efficiency, the fabrication cost of such units plays a crucial role for the realization of large area sensor networks.

In this study, we expand on the idea of performing an optimization of PANI films and determining suitable dopants and concentrations after analysis of the resulting sensitivities that come from material optimization. Furthermore, we investigate the statistical quality of sensor data resulting from doped and undoped sensors and the influence of the sensor material composition on the environmental stability against humidity changes.

## 2. Materials and Methods

Polyaniline (emeraldine base) (average MW = 5 kDa), carbon mesoporous (carbon black) (<500 nm particle size (DLS), >99.95% trace metals basis), *N*,*N*-dimethylformamide (DMF) (99.8% grade), ammonium hydroxide (28–30% aqueous solution, 99% purity), acetonitrile (99.9% grade), butane-2,3-dione (97% grade), ethanol (99.97% grade), and propan-2-ol (99.5% grade) were purchased from Merck, hexane (98% grade) was purchased from VWR, and 2,3,5,6-tetrafluoro-7,7,8,8-tetracyanoquinodimethane (F4TCNQ) was purchased from Ossila.

**Electrode design**: The printed circuit boards (PCB) used for the study and measurements were custom designed and ordered from Multi Circuit Boards Ltd. (Edinburgh, UK). The design consists of 32 single interdigitated electrodes with a channel length of 100 µm and width of 45 mm forming 16 pairs, where pairs are connected by means of sharing the ground electrode. Hence, a differential sensing principle was applied, measuring the resistance changes by comparison of a measurement and reference channel; one exposed to the sensing environment, while the reference sensor element was encapsulated. This approach allowed for the use of a wide range of CP materials in a chemiresistive configuration, while at the same time compensating the influence of temperature [23]. A single PCB board can hold an array of 16 sensor units, where each sensor unit consists of a measurement and reference electrode (see Appendix A). The reference electrodes were passivated with a silicon adhesive PLOFLON^®^ Teflon film SW10007. Therefore, a laser cutter (Master 2, 20 W, Ortur) was used at 100% intensity and 800 mm/s to match the specific shapes of the PCB layout and covered only the reference electrodes. The resulting covers were manually attached to the PCB.

**Polymer solution preparation**: Initially, polyaniline powder (MW = 5 kDa, CAS 25233-30-1) was weight and dissolved in DMF in a 15 mg/mL stock solution. Similarly, a stock solution (2 mg/mL) of the dopant, 2,3,5,6-tetrafluoro-7,7,8,8-tetracyanoquinodimethane (F4TCNQ, Ossila, 276.15 g/mol, CAS 29261-33-4) and carbon-black 10 mg/mL were also created in DMF. Carbon black solutions were additionally sonicated for 30 min at 65 °C with 1000 W to facilitate the dissolving (or dispersing) step. Subsequently, stock solutions were mixed to realize different PANI dopant weight ratios: 0.50, 1.00, 1.50, 2.00, 2.50, 3.00, 3.25, 3.50, 3.75, 4.00, 4.25, 4.50, 4.75, and 5.00 for F4TCNQ, and 0.5, 1.0, 1.5, 2.0, 2.5, 3.0, 3.5, and 4.0 for carbon black. For the gradient measurement of the relative humidity environment, the PANI solution, doped with carbon black, was created by adding hypromellose (4.5 mg/mL solvent) as a thickening agent. These solutions were heated at 65 °C and vortexed for 5 min to ensure proper mixing.

**Polymer deposition**: The pristine PCB boards were rinsed with 99.97% ethanol (Merck) for 1 min, dried with compressed air, cleaned using the UV-ozone process (FHR, UVOH 150 LAB) for 15 min and finally heat treated at 65 °C for 1 h on a hotplate. Next, 2.5 µL of the PANI dopant solutions were pipetted onto the measurement electrode of each sensor pair of the PCB channels, dried overnight, and measured separately with a Fluke (87 V True RMS) multimeter. Due to the parallel circuit of the PCB electrode design, and to prevent distortion of the measured resistance, the resistance was measured on the following day with a Fluke (87 V True RMS) multimeter separately for each channel. After that, the second interdigitated electrode of each sensor pair was covered with the same polymer solution in the same manner as on the day before and left to dry overnight. The resulting functional PCB boards were then stored for one week to ensure complete evaporation of any residual solvent within the film before being used for measurements. 

**Gas calibration measurements**: Titration measurements of ammonia were performed using a liquid calibration unit (LCU, Ionicon Analytik, Innsbruck, Austria) and transforming a known liquid concentration into a gas phase. To this end, a custom-made holder and gas chamber were connected to the gas outlet of the LCU, ensuring a continuous gas flow over the sensor boards for the duration of the measurements. The LCU was also equipped to an automated valve switch unit (VICI^®^ Universal Electric Actuator Model EUHB with a Cheminert^®^ Model C25Z stream selector) in order to switch through up to 14 different input channels with Tygon (Techlab, Blacksburg, VA, USA) tubings (diameter 0.76″) and PEEK (VICI^®^ Grooved PEEK Ferrules 1/16″, Vici High Pressure PEEK Nuts 1/16″ hex) connectors. Using this system, ammonia concentrations ranging from 0.01 to 50 ppm were generated by adjusting the liquid inlet flow rate (µL/min) and gas flow (SCCM) of the 0.45 M ammonia (28–30% ammonia hydroxide, Merck) feedstock solution.

Detailed calculations of the transfer from the liquid to the gas phase are shown in Appendix A. A baseline was established for 7 h while flushing ambient air at a rate of 500 SCCM and dispersing Milli-Q water in the gas phase to ensure constant relative humidity of 50% inside the gas chamber. Starting with the lowest concentration, each ammonia solution was purged through the LCU into the gas chamber for 3 h until the highest concentration was reached. To observe the recovery of the sensor, the system was purged with ambient air kept at a constant 50% relative humidity for 17 h. To perform analytical studies on the polymer composite baseline reactions to the change in relative humidity, a second input channel (inlet 2), connected to a reservoir of Di-H_2_O of the LCU, was utilized. Humidity gradients over the course of 18 h were established by a stepwise in, or decrease in, the water input channel flow rate for intervals of 5 min per step.

**Data logging**: Each polymer sensor signal was obtained with a multiplexer and a 16-bit analog-to-digital converter (LTC2497, Analog Devices, Norwood, MA, USA), resulting in a signal resolution of 65,535 quantification steps. Additionally, a humidity and temperature sensor (SHT85, Sensirion AG, Stefa, Switzerland) was installed adjacent to the polymer sensors. A microcontroller collected all sensors’ outputs sequentially, resulting in an update rate of 2.1 s for all 16 sensors. The latest sensor signals were transmitted to a PC every second for logging and monitoring in real time via a graphical user interface (GUI).

**Analyte measurements:** Ammonium hydroxide (28–30%) solution was diluted to a 0.150 mM concentration, hexane (98% grade), acetonitrile (99.8% grade), ethanol (99.97% grade), and propan-2-ol (99.5% grade) were used as is, and butane-2,3-dione (97% grade) was diluted in Di-H_2_O to a 0.2 µM solution. These solutions were then transferred to glass vials with an inner diameter of 50 mm, each holding 20 mL of the analyte solution. In each glass vial, two paper strips (paper test strips, 15 cm × 0.7 cm, SUPVOX) were inserted in such a manner that one end of the paper strip was in contact with the repository bottom and the other end protruded from the solution with a length of 1 cm. Each vial was positioned inside a plastic box (h/w/l = 15.5/8/13.8 cm) with a volume of 1.7 L. These boxes were covered with spring-hinged hatches, holding the containers air tight, but allowing for opening of the lids from outside; thereby, the partial pressure of the analyte in the gas phase inside the boxes approached equilibrium conditions over time. After an initial equilibration time, the above-described polymeric sensor system was introduced into the boxes, while marking the time stamp of insertion. The sensors were placed inside the analyte boxes for five minutes and then removed and placed in empty room atmosphere conditions for ten minutes before repeating the process with the subsequent analyte box. In such manner, the sensor was circled through all analyte-containing boxes yielding a 5-min exposure time, followed by 10 min of regeneration. This procedure was repeated 6 times for the doped PANI sensor array and 5 times for the pristine PANI sensor array for each analyte. Scanning electron micrographs were recorded with a Zeiss SUPRA 40 field emission scanning electron microscope.

## 3. Results

### 3.1. Resistance of Doped Polyaniline Films

PANI films, fabricated with different F4TCNQ doping rations ranging from 0.5% up to 5 weight percent (wt%), were fabricated as described in the materials section and their resistance was measured after one week of resting to ensure the complete evaporation of any residual solvents. This procedure was performed for eight channels of each dopant concentration, and the resulting resistances are shown as box plots in Figure 1a. A clear trend of decreasing resistance is observed as dopant concentrations increase. The lowest F4TCNQ concentration (0.5 wt%) resulted in an average film resistance of 14.8 ± 6.5 MΩ, while the highest concentration (5 wt%) resulted in 750 ± 184 kΩ.

Generally, an almost linear trend in the resistance decrease with increasing dopant concentration is observed. Interestingly, a small deviation from this trend can be seen between 4% and 4.25% (wt%) of F4TCNQ. We speculate that this is a statistical artifact, resulting from the variation in the measurement data. It can be seen that an addition of less than 5 wt% F4TCNQ did not result in any visual modification of the polyaniline films. Furthermore, another series of PANI films were prepared, but this time carbon black mesoporous powder was used as a dopant. Carbon black does act as a filler material, increasing the overall conductivity of the films not due to chemical interaction with the polymeric matrix, but filling gaps in the matrix and forming a percolation pathway for electrons through and around the polymer. Interestingly, carbon black did not form a homogenous solution with the polymer in DMF, instead it formed a suspension. The resulting mixture was applied directly after stirring. Same as with F4TCNQ, the resulting sensor surfaces were dried in air for a week before measuring the resistance (Figure 1b). Again, a trend for a decreasing resistance with increasing carbon black concentration was observed. However, the variance obtained for each carbon black concentration was higher than compared to F4TCNQ samples, indicating less homogenous solution and percolation pathways that were statistically dependent on the locally applied carbon black content on the electrode pair. Overall, the resulting resistances were significantly higher than for F4TCNQ, with a resistance of 76 ± 15 MΩ for 0.5 wt% carbon black and 38 ± 11 MΩ for 3.5 wt%. In contrast to previous studies where carbon black was used to increase the conductivity of waterborne polyurethane films [25], the reproducibility of these results suffers from a larger variance. This effect most likely results from the non-ideal conditions for carbon black solutes in DMF. Although some studies proposed that PANI and carbon black display synergistic behavior forming electrodes [26,27], these studies did not investigate the reproducibility of such composites and used significantly higher carbon black concentrations. As previously described, polymer carbon black composites, upon blending and processing, may undergo various processes, such as deagglomeration, aggregate erosion, fracturing, and re-agglomeration, resulting in a nonhomogeneous distribution of particles [28].

Furthermore, to investigate the surface morphologies, we analyzed the resulting surface via scanning electrode microscopy. Therefore, a PANI film doped with 2 wt% of carbon black was chosen. The resulting SEM image is shown in Figure 2. Scanning across the composite surface displayed several cavity-like structures with diameters in the sub-µm scale. The resulting structures are porous and allow diffusion of VOC and other gaseous particles into the bulk of the resulting composites. It is shown that the kinetic reaction time of nanostructured polymer films is faster for smaller polymeric structures and slower for more bulky surfaces [29]. Previous studies show similar approaches, using control of the solvent evaporation rate or relative humidity conditions to form porous semiconducting films for organic electronics, block copolymer membranes with ultrahigh porosity pore structures [30], or form porous poly (vinylidene fluoride-trifluoroethylene) [31,32,33]. However, to the best of our knowledge, no previous studies demonstrated the formation of porous PANI composite films to form surface structures suitable for smell sensing applications.

### 3.2. Responses to Analytes

#### 3.2.1. Liquid Calibration Unit Measurements

A titration of ammonia solutions, resulting in gas phase concentrations of 0.01, 0.5, 1, 5, 10, 25, and 50 ppm in a Liquid Calibration Unit (LCU), was performed for PANI sensors doped with F4TCNQ concentrations ranging from 3.25%, 3.5%, 4%, 4.5%, and 5% (wt%), and the results are shown in Figure 3. Simultaneously, the *y*-axis in Figure 3 was calculated from the differential principle of the sensor units, assuming that the reference channel (beforehand passivated) did not change during the experiment, to obtain the total resistance value, as shown in Figure 3. It can be seen that the initial sensor resistance is indeed in good agreement with the dopant concentration, as shown in Figure 1. At the same time, a decrease in response amplitude (sensitivity) is observed with increasing dopant concentrations. The dependence of the response on the dopant concentration is shown in Appendix A, where the responses were normalized as a percentage of the sensor resistance. For all sensor units, a similar time constant is observed in which the initial sensor response is almost instant, but the system requires about 15 min before reaching equilibrium conditions (Appendix A). Such slow kinetics clearly point at a diffusive adsorption process in the porous structures of the bulk sensing material, which agrees with the observations shown in Figure 2. Furthermore, the dependence of the sensor noise level on the dopant concentrations was analyzed, showing that the lowest doping levels exhibit the highest noise, with signal standard deviations of 5557, 3616, 1943, 924, and 661 ohm, measured over a time span of 10 min for dopant concentrations of 3.25%, 3.5%, 4%, 4.5%, and 5% (wt%), respectively. Due to the principle of data recording and the resolution of the sensor signals with a 16-bit analog-to-digital converter, the noise level of the highest dopant concentrations was not fully resolved. The results of the titration experiment show that, for an ammonia concentration of 100 ppb, the PANI sensors doped with 3.25%, 3.5%, and 4% (wt%) F4TCNQ exhibit a signal-to-noise ratio above three, hence the limit of detection (LoD) according to the IUPAC definition for these sensors can be estimated to be around 100 ppb ((∆I/I_0_)_mean_ ± kσ). The polyaniline sensors doped with 4.5% and 5% (wt%), on the other hand, showed a diminished reaction to such low NH_3_ concentrations, and the LoD was 500 ppb and 1 ppm, respectively. However, using a higher ADC bit depth resolution and an incremental ΣΔ modulator to resolve the dynamic part of the sensor signal could further decrease the LoD of the sensors with higher dopant levels [34]. The SNR will be improved due to increased signal resolution and the usage of more bits for the dynamic section of the signal, hence allowing for investigating minute polymer conductance changes [34]. A comparison to other sensor platforms using nanocrystalline zinc oxide sensors shows that the addition of aluminium as a dopant increases the sensitivity for hydrogen detection with an optimum increase for an addition of 3% [35]. Furthermore, other investigations show that CuO thin film sensors exhibit the highest sensitivity, with an addition of around 3% of Lanthanum in a sol–gel method fabrication process [36]. Interestingly, lower and higher dopant concentrations do not result in better sensor performance. The authors hypothesize that resulting grain sizes and surface area increases play a significant role for this effect. Moreover, the depletion region extends deeper in the bulk. However, with increasing bulk conductivity, a shift in depletion region thickness has less influence on the film conductivity, hence decreasing sensitivity. We contemplate that a trade-off of these factors results in a maximum sensitivity at a low dopant concentration for the above-mentioned studies. Furthermore, our study does not support these claims, since the addition of dopant concentrations in a PANI solute does not necessarily increase the surface area of the resulting films with the fabrication method described in this work. In previous work [37], we demonstrate the influence of heating temperature, and annealing parameters on PANI sensor sensitivity, which is in good agreement with the explanations cited above. Hence, the results shown in Figure 1 exhibit only one of the sensitivity scaling effects, with the highest sensor sensitivity for the lowest dopant concentration. To the best of our knowledge, no investigations on the sensitivity of PANI films for gas sensing in the dependence on dopant concentration were previously conducted, and similar studies on other material films do not include any information on the fabrication reproducibility of the dependence on the dopant concentration. Comparing the titration experiment results to literature, the lowest concentrations of ammonia detected were in the ppm regime for polyaniline films [38], 1 ppm for graphene/polyaniline composites [39], 1 ppm for acrylic acid-doped polyaniline [40], and 5 ppm to up to 100 ppm for polyaniline-based sensing elements, as summarized in a recent review by Tanguy et al. on polymer sensors for ammonia detection [41]. For PANI on a bacterial cellulose being used as a wearable sensor substrate with co-dopants of sulfosalicylic acid (SSA) and poly (2-acrylamido-2-methyl-1-propane sulfonic acid) (PAMP), a LoD of 10 ppb to ammonia can be reached [42]. PANI composites blended with PMMA/PS and MWCNT are found to have a LoD of 830 ppm when considering an ammonia detection [43]. A summary is shown in Table 1.

Evaluation of Figure 3 reveals that the sensors exhibit a baseline drift that correlates to the analyte concentration, which is most prominent for the lowest dopant concentrations and highest analyte concentrations. Further measurements for the clear demonstration of this effect are shown in the Appendix A. Furthermore, this observation was correlated to a theoretical framework, considering the relevant gas laws. According to Henry’s law, the amount of dissolved gas in a liquid is proportional to the partial pressure in the gas phase. The partial pressure in the gas phase can be directly calculated from the analyte concentration in the gas phase (given in ppm). Since the PANI films have a very high surface roughness with cavities in the nm regime (see Figure 2), we can describe the PANI film as a porous material, where a similar behavior is observed as stated by Henry’s law. Hence, we denote the proportionality constant correlating the environmental partial gas pressure with the sensor baseline drift as the Henry constant.

For the description of the sensor response to each separate concentration injection, a simple exponential equation was used to extract features from the sensor kinetics:ΔResponse = A·e^−x(t − t^_0_^)^ + H·(t − t_0_) (1)
where ΔResponse is the change in the differential measurement curve over the time frame of the exposure to the analyte concentration; A is the amplitude of the sensor response; −x is the kinetic time constant; and H (Henry constant) is a constant which correlates with the baseline drift and the time t. The resulting model system fits very well with the sensor responses shown in Figure 3 and Appendix A. Generally, the model accuracy was found with an R2 above 0.95. Especially for very high analyte concentrations, the model is still capable of describing the sensor output accurately, as shown in Appendix A.

#### 3.2.2. Closed Volume Measurements and Statistical Evaluation

We also investigated sensor responses to a wider variety of relevant analytes, such as acetonitrile, ammonia, butane-2,3-dione, ethanol, hexane, and propan-2-ol, using a sensor array of 16 sensor units using 4.25 wt% F4TCNQ doped PANI films. These analytes were chosen as they are identified as some of the most significant lung cancer markers in exhaled human breath [44]. For these analytes, a headspace sampling method (1.7 L) was utilized. From the titration shown in Figure 3, a calibration curve was obtained and used to determine the ammonia concentration corresponding to the sensor signal responses in the headspace sampling method. Following this procedure, the ammonia concentration inside the pre-sampling chamber can be estimated to be around 3 ppm, as the ΔResponse of the sensor (change in baseline level upon exposure after 300 s) was around 4% for that concentration. The volatility (vapor pressure) of the chemical compounds is the main defining factor for differences in the concentrations, since the geometry, surface area, and evaporation time for all analytes were kept constant. The evaporation rate of the chemicals were calculated according to the procedure proposed by Mackay and Wesenbeeck [45], and is approximately 2.5 µg cm^−2^ h^−1^ for ammonium hydroxide. It was found that 3 ppm of ammonia in the used volume of 1.7 L corresponds to a total mass of 2.25 µg ammonia in the gas phase, which leads to an evaporation time of one hour, which is in good agreement with the result obtained above from comparison to the calibration curve. Calculations of the ppm of all other analytes are shown in Appendix A. However, the analyte sensor material interaction depends highly on the adsorption affinity and the diffusion parameters, which are highly specific for each molecule.

Figure 4 shows histograms of sensor responses to all six analytes, where F4TCNQ-doped sensors are shown as red bars and pristine PANI sensors as black bars. The sensor response kinetics for all analytes, including the change in humidity and temperature upon exposure, are shown in the Appendix A. A clearly different sensor behavior is observed when comparing doped and undoped F4TCNQ sensors, including changes of the magnitude of the sensor response, the kinetics, baseline drift, and the affinity for specific analytes. The differences are shown as radar plots in Appendix A. For all analytes, a good approximation to Gaussian distribution of the sensor signal can be seen. Appendix A shows the average response amplitudes of pristine and doped PANI sensors for all six analytes. In general, a higher sensor response is observed for F4TCNQ-doped polyaniline sensors, with the exception of butane-2,3-dione. Interestingly, it is the only compound resulting in a sensor resistance increase, thus indicating a redox reaction of the analyte to the polyaniline bulk. As F4TCNQ is a *p*-type dopant, this reaction is decreased with less reactive electrons due to the dopant resulting in the lower response signals. The data of the sensor responses of F4TCNQ-doped and pristine PANI was further evaluated by principal component analysis. To this end, the datasets of two sensor units, including the ΔResponse value after 300 s, the fitted kinetic time constant k in that time frame, and the Henry constants of these time intervals were used as features. Hence, for each dataset, six independent features were observed and the sensor responses from other devices labeled as repetitions. The features were then standard scaled by a *sklearn* toolkit standard scaler for Python environments to normalize for the mean value and the variance for all the extracted features. As a next step, the six normalized features were reduced in dimension by principal component analysis (PCA) down to two dimensions. The process was again carried out by a *sklearn* toolkit in Python. The same procedure was applied for the datasets of the F4TCNQ-doped and pristine polyaniline composite films. Principal component analysis generally can be used for dimensionality reduction in data pre-processing for machine learning applications [46]. Furthermore, in this work, we use it as a demonstration and visual representation for analyte differentiation even after drastic dimensionality reduction of the extracted features. Figure 5 shows the resulting classification results.

From the distribution of data points in Figure 5, again a clear difference between pristine PANI (b) and F4TCNQ-doped PANI films (a) can be observed. Although the dataset for the doped sensors was slightly bigger, in contrast to pristine PANI sensors, almost no overlaps in the principal component representation are observed for the clusters of ammonia and acetonitrile. Additionally, for the tighter packed data clusters of ethanol, hexane, and propan-2-ol, which all had significantly lower sensor responses, a distinction of the chemical classes is much better when using doped PANI films since the principal components of ethanol are less scattered.

### 3.3. PANI Conductivity Changes with Relative Humidity

For the application of polymer gas sensors in real-world conditions, the sensor signal dependence on the environmental conditions is of paramount importance. Apart from the sensor response, a hysteresis behavior of the sensor system can also be observed for polymeric sensors [47]. Previous studies show that polymer-based humidity sensors suffer from large hysteresis between absorption and desorption of moisture, which is a challenge, since it may lead to deformation of the polymer and thus affect the sensor performance adversely [48]. To evaluate the sensor dependence of pristine PANI, carbon black-doped polyaniline, and F4TCNQ-doped polyaniline, a gradient of the relative humidity environment was performed as described in the Materials section. The sensors were exposed for 8.5 h to an increasing RH from 10% up to 90%, followed by a decrease in RH from 90% down to 10% again. To achieve this, the flow rate of the second channel on the LCU was incrementally adjusted so that a gradient of relative humidity (RH) was achieved. From these curves, the humidity vs. sensor response curves were plotted for the increasing and decreasing regime of RH, and the result is shown in Figure 6.

## 4. Discussion

The herein investigated system exhibits a myriad of possible reactions and effects that ultimately result in a sensor signal. Most notably PANI is a *p*-type conductive polymer that can undergo protonic and/or redox doping, which heavily affects its conductivity [5]. Additionally, polymer swelling due to humidity variations also affects the electrical properties. Finally, the employed electrophilic F4TCNQ dopant is susceptible to nucleophilic attacks from amines such as ammonia [49]. It is not within the scope of this work to unravel this complex interplay of effects and yield comprehensive mechanistic insights, but rather, we report on reproducibility-driven fabrication optimization towards improved sensor performances. Protonic doping of the emeraldine base form of PANI allows for reversible transition to its reduced, and thus significantly higher, conducting (up to 10 order of magnitude) emeraldine salt form [50], which is depicted in Figure 7. Note that the different nitrogen configurations in PANI, with amine (-NH-) and imine (=N-) moieties will result in different sensing properties. For instance, when PANI is exposed to ammonia gas, it undergoes reversible dedoping by protonation (reduction of conductivity) [38,51]. Simultaneously, this effect can be enhanced as the NH_3_ also interacts with the F4TCNQ via nucleophilic attack, which results in the dedoping of F4TCNQ, thus further decreasing resistance (see Appendix A). The heteroatom N of the polyaniline contributes stronger to the conjugation of the polymer backbone compared to the heteroatoms of other common conductive polymers, e.g., polythiophenes and polypyrroles. The nitrogen atom of the polyaniline is regarded as a separate component B of the A-B polymer configuration. In contrast, polythiophenes and polypyrroles are of an A-A configuration type [5]. In Figure 7, the protonation of the emeraldine base form of polyaniline with amine (-NH-) and imine (=N-) compounds in equal proportions to the conductive (polaron) emeraldine salt is shown [52]. Since the emeraldine salt form of polyaniline is conductive (while leucoemeraldine and pernigraniline are not) doping with the very prominent *p*-type dopant F4TCNQ results in this structure, where both anions A^-^ represent the double negatively charged F4TCNQ molecular dopant [53].

The sensor response towards ethanol and acetonitrile could be explained via their high electric constants that promote strong interactions with the nitrogen atoms, thus leading to an expansion of the PANI chains into more stretched conformations—secondary doping [51,54]. According to Bai et al. [51], this is expected to increase the crystallinity of the polymer and decrease its electrical resistance. Our data partially supports this claim in the case of ethanol (Appendix A), but once doped (Appendix A) a different mechanism must be at play. Other studies investigated the sensing mechanism of a nanomaterial-based sensor towards health-hazardous gases, with low LoDs and high stability, indicating a promising acetone sensor [55].

We also observe a porous surface structure and low analyte adhesion times, indicating diffusive effects as the main source of signal transduction. For all analytes tested in the experimental section, we observe a similar behavior, except for butane-2,3-dione. The opposite response behavior for butane-2,3-dione, as shown in Appendix A, is indicated in the assumed reaction mechanisms, shown in Figure 7, for ammonia and butane-2,3-dione [51]. While the interaction of PANI with ammonia decreases the conductivity, butane-2,3-dione leads to an increased conductivity and response, as observed in our studies. The observed high sensitivity with LoDs in the ppb regime and the slow adsorption kinetic time scales are in good agreement with previous studies on nanostructured polymer films for gas sensing [29]. The presented sensor system indeed has a high surface roughness and area (high sensitivity), as well as thick and large material film due to the printing technique (slow time constants). Moreover, the above-described mechanisms based on polaron transport, oxidative states, and reduction, are the main interactions with the environment. Due to the described observation of the kinetics of interactions with the material, these results suggest a low surface-to-bulk ratio for doped polyaniline films fabricated by the method in this work.

## 5. Conclusions

We demonstrated the successful fabrication of polyaniline films in its pristine state and doped with carbon black and F4TCNQ. The doping with F4TCNQ was found to be more reproducible and yielded superior sensing performance. In titration experiments, the lowest LoD found for ammonium hydroxide in the gas phase measurements using F4TCNQ-doped sensors was 100 ppb, which is lower than previous reports of PANI ammonia sensors. We speculate that the LoD could be further lowered by the tuning of additional parameters, such as higher bit depth, thinner substrate thickness, smaller electrode dimensions, and three-dimensional electrode surfaces. Furthermore, we correlate the response curves of PANI sensors to single component interactions using a simple equation with a good fitting accuracy of R2 > 95% and explain the sensor drift with a diffusion model via a Henry constant. Decreased sensor signals were observed for higher dopant concentrations. Statistical evaluation of the sensor responses revealed a Gaussian distribution for sensing experiment reproducibility and good reversibility, while both these criteria were significantly better for PANI films doped with F4TCNQ than undoped PANI films. Hence, we conclude that a trade-off between sensitivity and reproducibility has to be established when choosing dopant levels for PANI films for gas sensor applications. Environmental stability and reversibility upon variations of the environmental humidity were significantly improved for the F4TCNQ-doped PANI sensors compared to pristine or carbon black-doped PANI. With regard to applications of the sensor material for eNose systems, we found from principal component analysis that PANI sensors doped with 4.25 wt% F4TCNQ can discriminate between at least five different cancer marker molecules (3 ppm) with good accuracy.

## Figures and Tables

**Figure 1 sensors-22-05379-f001:**
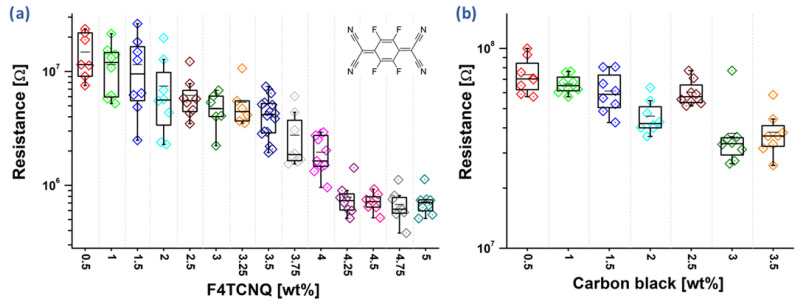
(**a**) Box plots of F4TCNQ-doped polyaniline sensor units, the insert shows the chemical structure of F4TCNQ; (**b**) box plot of carbon black-doped polyaniline sensor units.

**Figure 2 sensors-22-05379-f002:**
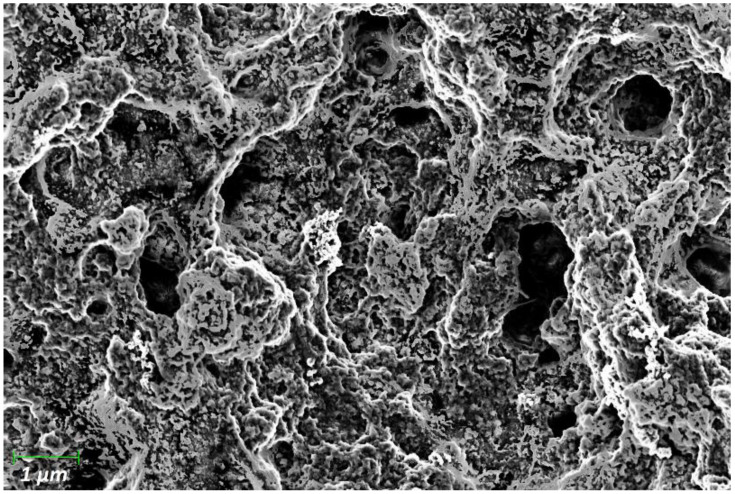
Scanning an electron microscope image of a polyaniline film doped with 2 wt% carbon black.

**Figure 3 sensors-22-05379-f003:**
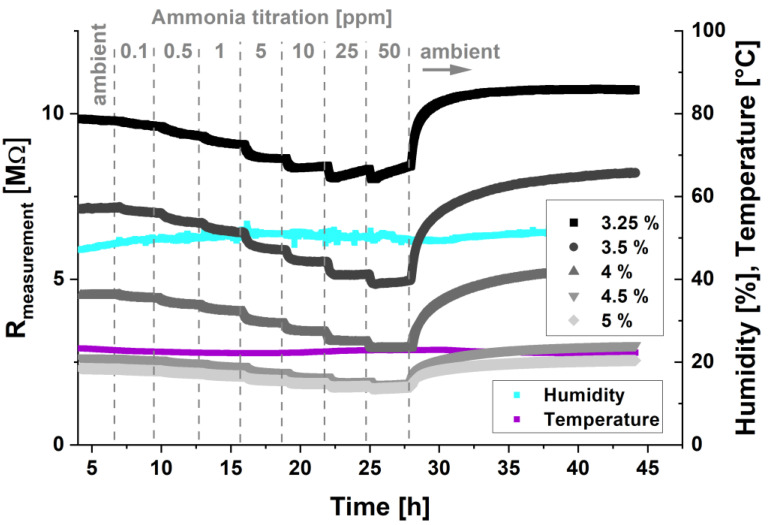
Liquid calibration unit measurements of polyaniline sensors doped with different F4TCNQ concentrations—3.25% (black curve), 3.5% (red curve), 4% (blue curve), 4.5% (orange curve), and 5% (wt%) (green curve). Humidity (turquoise) and temperature (purple) were monitored during the whole duration of the experiments.

**Figure 4 sensors-22-05379-f004:**
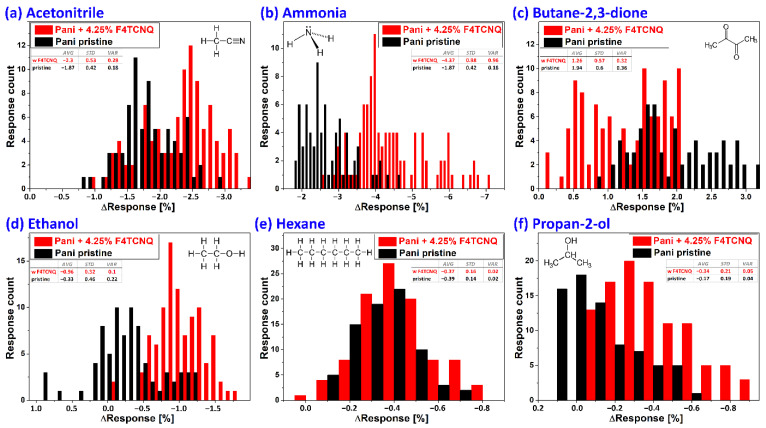
Histogram (bin sizes 0.1%) comparison of sensor responses to six different analytes with pristine polyaniline films (black bars) and PANI films doped with 4.25 wt% F4TCNQ (red bars). (**a**) Acetonitrile responses; (**b**) ammonia; (**c**) butane-2,3-dione; (**d**) ethanol; (**e**) hexane; and (**f**) propan-2-ol.

**Figure 5 sensors-22-05379-f005:**
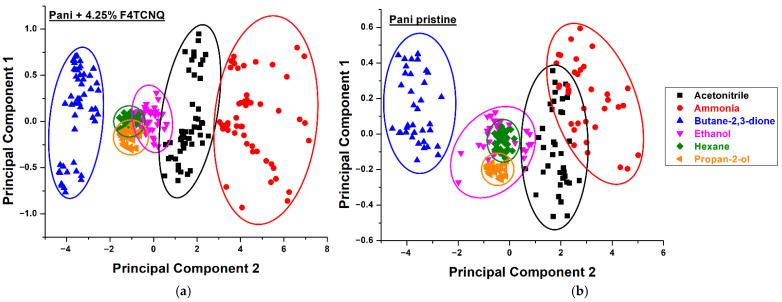
PCA for six different analyte gases measured with (**a**) eight F4TCNQ-doped polyaniline sensors and (**b**) six undoped polyaniline sensors.

**Figure 6 sensors-22-05379-f006:**
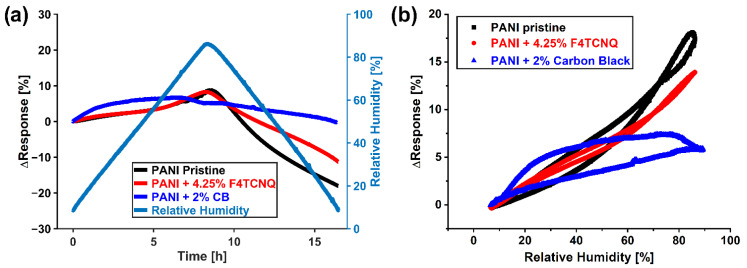
(**a**) Baseline changes of pristine (black), F4TCNQ-doped (red), and carbon black-doped (blue) sensors when exposed to a gradient of environmental humidity measured in a liquid calibration unit and (**b**) resulting hysteresis curves for the respective sensors.

**Figure 7 sensors-22-05379-f007:**
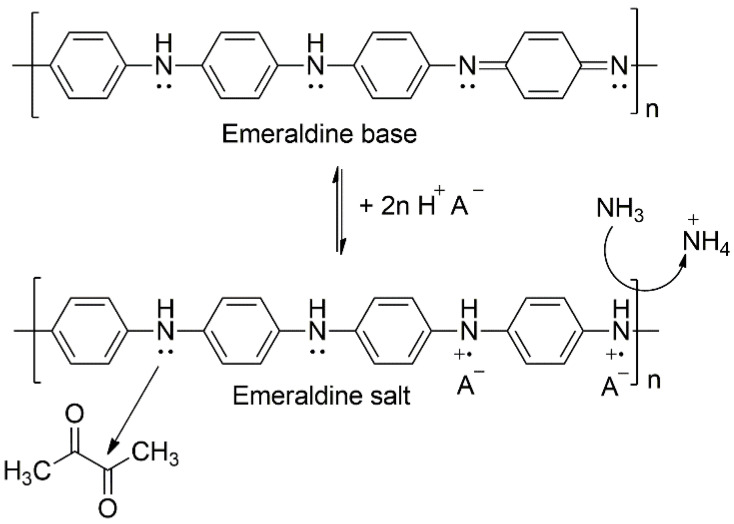
Chemical structures of the emeraldine base (**top**) and the protonated salt form (**bottom**). Possible interaction of the PANI nitrogen with butane-2,3-dione and ammonia are shown.

**Table 1 sensors-22-05379-t001:** Performance of various PANI based devices regarding ammonia detection.

Dopant	NH_3_ LoD (ppm)	Reference
Graphene	1	[39]
Acrylic acid	1	[40]
None	5–100	[41]
SSA and PAMP	0.01	[42]
MWCNT	0.83	[43]
F4TCNQ	0.1	This work

## Data Availability

Data available in a publicly accessible repository. The data presented in this study are openly available in [Zenodo.org] at [10.5281/zenodo.6857040].

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
