# Peer review of "Optimization of Printed Polyaniline Composites for Gas Sensing Applications"

_sensors, 2022, doi:10.3390/s22145379_

Round 1

Reviewer 1 Report

This article investigates the sensitivity and specificity of two different dopant concentrations (F4TCNQ and carbon black) for five relevant markers of respiratory cancer diagnosis. The research results can guide the development of polyaniline thin-film gas sensors. The structure of the article is clear. The article describes the experimental steps in great detail. The experimental content can illustrate the conclusion of the paper. I think the paper gives very good work and is acceptable for publication. There are two suggestions:

1. To modify Figure 3 and Figure 6. At present, there are many overlapping points, and some points can be omitted.

2. To clarify the relationship between principal component analysis and machine learning.

Author Response

We would like to thank the reviewer for the constructive input. The original Reviewer suggestions are colored dark blue and numbered by the paragraphs originally set by the reviewers. Our replies are colored black. Please see the attached word file for the replies

Reviewer 2 Report

R.-Rozman et al. studied a research work entitled "Optimization of printed polyaniline composites for gas sensing applications". The manuscript is well written and exciting for readers. However, a few critical points need to be addressed before publishing the paper. Therefore, I recommend a minor revision for the publication in the Sensors.

I strongly encourage the authors to revise and consider this point carefully:

1)        The author should split the introduction into two paragraphs and also add 3rd paragraph for the following sentences "In this study we expand on the idea of performing an optimization of PANI films and determining suitable dopants and concentrations after analysis of the resulting sensitivities as a result of material optimization. Furthermore, we investigate the statistical quality of sensor data resulting from doped and undoped sensors and the influence of the sensor material composition on the environmental stability against humidity changes". It will be good for readers.

2)        The authors should correct the typo errors in the references.

3)        The author stated that "at the same time a decrease of response amplitude (sensitivity) is observed with increasing dopant concentrations" in lines 249-250. The author should discuss the possible reason with references.

4)        The author stated that “The results of titration experiment show that for an ammonia concentration of 100 ppb the PANI sensors doped with 3.25%, 3.5% and 4% (wt%) F4TCNQ exhibit a signal to noise ratio above 3, hence the Limit of Detection (LoD) according to the IUPAC definition for these sensors can be estimated to be around 100 ppb ((ΔI/I0)mean ± kσ. The polyaniline sensors doped with 4.5% and 5% (wt%) on the other hand showed a diminished reaction to such low NH3 concentrations and the LoD was 500 ppb and 1 ppm respectively. However, using a higher bit depth resolution of the signal data most likely would further decrease the LoD of the sensors with higher dopant levels in lines 264-271. The explanation is missing, and the author should discuss with references.

5)        The author should also discuss the following reference in the discussion part of the revised manuscript; Micromachines 2021, 12(6), 598; https://doi.org/10.3390/mi12060598.

Author Response

We would like to thank the reviewer for the constructive input. The original Reviewer suggestions are colored dark blue and numbered by the paragraphs originally set by the reviewers. Our replies are colored black. Please see the attached word file for our replies.

Reviewer 3 Report

(Line 239) Correct the number. Instead of the specified 1.2, it will be correct - 3.2. Responses to analytes

(Line 240) Correct the number. Instead of the specified 3.1.1, it will be correct - 3.2.1. Liquid calibration unit measurements

(Line 309) Correct the number. Instead of the specified 3.1.1, it will be correct - 3.2.2. Closed volume measurements and statistical evaluation

(Line 374) Correct the number. Instead of the specified 1.3, it will be correct - 3.3. PANI conductivity changes with relative humidity 

Author Response

We would like to thank the reviewer for the constructive input. We thank the reviewer for proposing an optimized numbering. We have adapted the numbering according to the suggestions for all lines labelled below.

Comments and Suggestions for Authors

(Line 239) Correct the number. Instead of the specified 1.2, it will be correct - 3.2. Responses to analytes

 (Line 240) Correct the number. Instead of the specified 3.1.1, it will be correct - 3.2.1. Liquid calibration unit measurements

 (Line 309) Correct the number. Instead of the specified 3.1.1, it will be correct - 3.2.2. Closed volume measurements and statistical evaluation

 (Line 374) Correct the number. Instead of the specified 1.3, it will be correct - 3.3. PANI conductivity changes with relative humidity

All discussion points suggested by the reviewers have been shared with all authors and all authors agree to any changes made in the revision process.

Kind regards,

On behalf of the team,
Ciril Reiner-Rozman